# Socio-Economic Factors Related to Drinking Water Source and Sanitation in Malaysia

**DOI:** 10.3390/ijerph17217933

**Published:** 2020-10-29

**Authors:** Yuke-Lin Kong, Jailani Anis-Syakira, Weng Hong Fun, Nur Zahirah Balqis-Ali, M. S. Shakirah, Sondi Sararaks

**Affiliations:** Health Outcomes Research Division, Institute for Health Systems Research, National Institutes of Health Malaysia, Ministry of Health Malaysia, Jalan Setia Murni U13/52, Seksyen U13 Setia Alam, Shah Alam 40170, Malaysia; anissyakira.j@moh.gov.my (J.A.-S.); fun.wh@moh.gov.my (W.H.F.); dr.nurbalqis@moh.gov.my (N.Z.B.-A.); shakirah.ms@moh.gov.my (M.S.S.); sararaks.s@moh.gov.my (S.S.)

**Keywords:** sanitation, drinking water source, house type, strata, income, education level

## Abstract

Access to improved water and sanitation is essential. We describe these practices in Malaysia using data from a nationwide community survey and used logistic regression to assess the determinants. Of the 7978 living quarters (LQs), 58.3% were in urban areas. About 2.4%, 0.5% and 27.4% of LQs had non-improved water sources, non-improved toilet types and improper domestic waste disposal, respectively. Open burning was practiced by 26.1%. Water source was a problem for long houses (10.5%), squatters (8.5%) and shared houses (4.0%). Non-improved toilet types were 11.9% for squatters and 4.8% for shared houses. Improper domestic waste disposal practices were higher for occupants of village houses (64.2%), long houses (54.4%), single houses (45.8%) and squatters (35.6%). An increase in education or income level was associated with a decrease in improper domestic waste disposal methods. House type significantly affected water and sanitation after adjusting for the effects of other variables. Lower household income was associated with non-improved toilet types and improper domestic waste disposal. Lower education and rural location influenced domestic waste disposal. The water and toilet facilities in Malaysia were generally good, while domestic waste management practices could be improved. There remain pockets of communities with environmental challenges for the nation.

## 1. Introduction

Access to water and sanitation, a basic foundation of public health, was acknowledged as a human right by the United Nations General Assembly a decade ago [1], reinforcing the fundamental role of clean water and adequate sanitation in maintaining good public health. The World Health Organisation (WHO) defines sanitation as the “provision of facilities and services for the safe disposal of human urine and faeces” [2]. This encompasses access to a safe system that considers a separation between human waste and human contact at all steps from waste collection, treatment and disposal. In maintaining good hygiene, access to proper domestic waste collection and disposal is of equal importance [2]. 

The WHO estimated that the annual improper sanitation deaths amount to 432,000 in low- and middle-income countries [3]. Improvement in water, sanitation and hygiene could prevent up to 297,000 under-5 mortalities annually [3]. The importance of safe water and sanitation was translated into the Sustainable Development Goal 6, to ensure access to water and sanitation for all (UN) [4], a global commitment in reducing disparities towards ensuring fulfilment of basic human rights. Access to drinking water services has reached 1.6 billion people [5] and basic sanitation reached 2.1 billion people [6] since 2000. Despite countries achieving better access to water and sanitation, gaps continue to exist within countries [7]. The 2017 Joint Monitoring Programme (JMP) report described a 2% drop in access to safely managed water between 2000 and 2015 in Malaysia, with rural populations affected more [8]. 

The conditions that people live in affect their access to clean water and proper sanitation. Despite the variable contexts in different countries, the common predictors of access to clean water and sanitation are income, education and geographical location [9,10,11,12,13,14]. The presence of a female head in an African household was associated with better access to clean water as women played a traditional role in fetching water for the household [9,10]. Temperature determined improved sanitation as lower temperatures in different geographical regions affected sanitation systems in China [13]. In Malaysia, water and sanitation is a challenge to be addressed among subgroups. Maintenance of installed water systems in indigenous population resettlements and rural areas were under the purview of local management who may not have adequate resources to ensure proper maintenance in provision of clean water [15].

Parallel to water and sanitation facility access, domestic waste management is an equally important component of hygiene that impacts health. A study in Nigeria found adverse health effects stemming from improper domestic waste management [16]. The World Bank reported a large divide in domestic waste management in 2016, where the percentage of waste collection in low- to middle-income countries were less than the average global rates [17]. Hygienic practices remain a challenge within the less affluent population due to their living conditions [18]. Hazardous domestic waste management practices were more prevalent among vulnerable populations, including the poor and minority groups [19].

Inequities in exposure to environmental risks leave the less affluent in a vulnerable position. In order to achieve SDG 6, it is important to explore access to clean water supply, sanitation facilities and domestic waste disposal in Malaysia, to implement strategies and policies for improvement. We seek to explore the socio-economic factors associated with non-improved drinking water sources and sanitation in Malaysia.

## 2. Materials and Methods

### 2.1. Data Source

We used data from the 2015 National Health and Morbidity Survey (NHMS) a nationwide cross-sectional community survey. It was implemented via a two-stage stratified random sampling design, by states and urban/rural localities, to yield a representative sample for the country. The detailed methodology of the survey is described elsewhere [20]. The study covered a total of 7978 living quarters (LQs), of which 4655 were from urban areas and 3323 from rural areas. The unit of analysis was LQs.

We analysed three main sanitation outcomes, namely, potable drinking water sources, toilet types and domestic waste disposal methods. We used the WHO and UNICEF JMP definitions of improved and non-improved drinking water sources and toilet types used [21]. Non-improved drinking water sources include water from unprotected dug wells, unprotected springs, surface water (river, dam, lake, pond, stream, canal and irrigation channel), vendor-provided water (cart with small tank/drum, or a tanker truck) and bottled water [21]. Bottled water is considered non-improved due to potential limitation of quantity rather than quality. Public or shared latrines, open pit latrines and bucket latrines are considered as non-improved facilities [21]. Domestic waste disposal covered in this study included only waste generated from households. Other wastes [22] from office, commercial area, and street waste were excluded.

### 2.2. Data Analysis

Cross tabulations were run between the types of water sources, toilets and domestic waste disposal used by the LQs with the demographic characteristics of the household heads. The socio-economic factors included were household income quintiles, where Q1 is the poorest 20% and Q5 is the richest 20% of the population; education level; geographical location; and house type. The data was processed using Epi Info software, version 21, and analysed using Statistical Package for the Social Sciences (SPSS) (IBM Corp., Armonk, NY, USA), version 22. 

Descriptive analysis was conducted to determine the relationship between sanitation with each socio-economic variable. Binary logistic regression was performed to identify the factors associated with non-improved water sources, non-improved toilet types and improper domestic waste disposal. The enter method was used in the logistic regression analysis and the odds ratio (OR) with 95% confidence intervals of the OR (CI) reported.

## 3. Results

Of the 7978 LQs, 58.3% were in urban areas. All house types were seen in both urban and rural areas, including long houses and village houses (Figure 1).

Almost all LQs had improved water sources and toilet types, while two thirds of the LQs had proper domestic waste disposal (Figure 2). A common improved water source was piped water into the house (89%), while the frequently found improved toilet type was a flush toilet connected to a main sewerage system (48.0%). In turn, the common domestic waste disposal method was local authority/management collection (65.7%) (Appendix A, Appendix B, Appendix C). 

About 2.4%, 0.5% and 27.4% of LQs had non-improved drinking water sources, non-improved toilet facilities and improper domestic waste disposal, respectively (Figure 2). Open burning was practiced by 26.1% of the LQs. A small percentage of LQs (4.6%) reported recycling of their domestic waste (Appendix C). 

No differences were observed between urban and rural locations for water sources and toilet types. However, improper domestic waste disposal methods, mainly open burning, was higher in rural LQs (51.9%) (Figure 3). Water sources was a problem for long houses (10.5%), squatters (8.5%) and shared houses (4.0%) (Figure 4). Non-improved toilet types were 11.9% for squatters and 4.8% for shared houses (Figure 4). Improper domestic waste disposal practices were higher in village houses (64.2%), long houses (54.4%), single houses (45.8%) and squatters (35.6%) (Figure 4). An increase in education or income levels showed a decrease in improper domestic waste disposal methods (Figure 5 and Figure 6).

House type significantly affected the drinking water source, toilet type and domestic waste disposal after adjusting for the effect of education level, household income and geographical location (Table 1). In addition, lower household income was associated with non-improved toilet types and improper domestic waste disposal. Lower education and rural location influenced domestic waste disposal. People living in squatter settlements had a higher likelihood of using a non-improved water source and non-improved toilet type, with an OR of 6.38 (95% CI: 2.39–17.04) and OR of 24.86 (95% CI: 9.15–67.52), respectively, as compared to single houses. For domestic waste disposal, a village house was more likely (OR 1.80, 95% CI: 1.56–2.09) to practice improper disposal as compared to a single house (Table 1). 

## 4. Discussion

Access to improved drinking water sources was available for almost all people living in Malaysia. Almost all LQs had improved toilet facilities, but many among squatters, long houses and village houses dwellers still used pour flush toilets. In terms of domestic waste disposal, the majority were regularly collected by the local authorities. Lack of proper domestic waste disposal was seen among village houses with open burning still commonly practiced and dumping into improper channels by squatters and long house residents. Compared with water sources and toilet types, only waste disposal methods were significantly affected by both education and income levels. 

Access to improved water sources is a crucial factor for a country’s sustainable growth and development. In Malaysia, long standing approaches by the governing bodies, beginning with the building of dams and water treatment facilities in the 1900s up to the supply of piped water in major towns from the 1930s to the 1970s, have improved this access [23]. The water sector operationalization has gone through various developments, including privatization of water handling companies in the 1990s, which saw mixed results. Some of the challenges encountered were (1) operational inefficiency; (2) ineffective governance and regulation; (3) budgetary constraints; and (4) poor environmental performance. A major reform was put in place in 2006 to address all the identified issues by regulating the existing policies, financing methods and infrastructure developments [23]. As a result, the coverage increased from 85% in 1990 to 95% in 2010 [24]. The availability of improved water sources in this study was 95.6%, similar to the reported figure by the UN-Water Global Analysis and Assessment of Sanitation and Drinking-Water (GLAAS) (96%) [25] and the WHO/UNICEF (97.0%) [26]. 

Although very small in numbers, this study showed dwellers of squatters and long houses were less likely to access improved water sources; hence, efforts to improve the coverage of water supply for these marginalised communities are needed. Similarly, the study also found communities in squatter areas, village areas and long houses had less satisfactory sanitation facilities. This is also seen globally where there is still a significant number of those who lacked sanitation services, using either open defaecation (892 million) or non-improved facilities, such as pit latrines without a platform, hanging or bucket latrines (856 million) [8]. In Malaysia, attention is needed to address the LQs still using non-improved sanitation types, such as bore hole latrine without a cover, bucket latrines and hanging latrines, and especially those without any form of latrine facilities. Addressing this issue of poor sanitation facilities will help in the reduction of waterborne diseases [27]. The results also showed that, among the same communities (squatters, village houses and long houses), there were still many who did not have access to proper service of domestic waste disposal. 

The challenges of these specific house types can be traced back to the chronological and historical development of Malaysia’s water, sanitation and solid waste management. Prior to the 1970s, Malays and Indians were living generally in rural areas, while the Chinese lived in urban areas [28]. In 1970, the New Economic Policies were introduced to encourage socioeconomic improvement and reduce the gap across ethnicities. As such, many, especially Malays, migrated to the urban areas, resulting in a rapid need for low-cost housing and opening of squatter settlements. There were major infrastructure and environmental concerns, including potable water supply and proper sanitation for these settlements, since the houses were not built in accordance with the approved specifications, and made worse by the ad-hoc manner of squatter establishments [28,29]. Contributing factors to the worse slum management included limited connectivity to piped water due to the haphazard nature of squatter settlements, exclusion from regional planning, and lack of formal property rights [30]. There was a lack of proper domestic waste disposal in squatter areas for the same reasons. In Kuala Lumpur, for example, open dumping of waste has been the main practice among squatters for domestic waste, often leading to pollution of groundwater and land surface since the dumpsites have no proper measure to control rainfall and run-off [31]. Various other challenges, including the poor quality of the low-cost flats built to relocate squatter dwellers, hindered the process of relocating and reducing the squatter settlements [32].

Similarly, the rapid urbanisation process saw most efforts to conserve the environment concentrated in bigger cities, leaving many rural areas with village houses lagging behind. For example, the water supply and establishment of pour flush latrines in Sarawak, one of the states in Malaysia with large rural areas and many long houses, only began in 1967 following the introduction of the Rural Health Improvement Scheme. The coverage of pour flush latrines increased steadily following the program’s introduction, from 44% in 1980 to 97% in 2002 [33].

The waste disposal service in rural areas was another area requiring regulatory improvement. Current regular collection is subjected to inclusion of an area under the management of either the Municipal Council, District Council, City Hall or Town Board, which leaves many rural areas not covered by such a service [34], leading to unsanitary waste management practices and disposal methods, such as open burning or throwing waste into drains, the river or sea, increasing the risk of air and water contamination. This lopsided service provision between urban and rural areas were also seen in other studies [10,35,36].

Despite various studies showing that the level of education and income affect access to water supply, proper sanitation, and waste disposal [12,37,38], this study found that only the waste disposal method was affected by both the education and household income of the heads of the households. As discussed earlier, the waste disposal method was heavily influenced by the house type and location of the house, whereby residents in rural areas not receiving a regular waste disposal service were forced to resort to other alternatives. However, higher education and awareness on the impact of disposal type on the environment could lead to residents choosing safer, more environmentally friendly methods as a disposal mechanism. This is in line with another study showing community education and awareness campaigns led to changes in disposal behaviour [39]. The alternative option, such as establishment of a community waste management unit, collection and transportation of garbage, may incur additional cost for residents.

Another area worth focusing on is domestic waste recycling, whereby the rate reported in Malaysia was 4.6% in comparison to Singapore’s 21% in 2016 [40], while Thailand recycled 11% of their domestic waste in 2003 [41]. This relatively low rate of recycling in Malaysia reflects the level of commitment towards environmental responsibility, thus potentially depleting the natural resources and increasing the need for opening more landfills [42]. 

For policymakers, to ensure universal access to improved water source and sanitation, attention is needed to address the disadvantaged communities. Addressing the needs of these communities would be beneficial in tackling various problems, including waterborne diseases, environmental concerns, public health expenditures, social development as well as the general wellbeing of the population [36]. 

In view of waste separation for recycling still being in its infancy in Malaysia, concentrated efforts to increase recycling practices are required. An appropriate recycling program could be introduced within an established standard operation to change the current attitudes and behaviours towards the habit of waste separation at the source, sustaining the momentum and continuous participation, rather than limited to relatively brief periods of implementation [43].

The data used in this study were from a nationwide community survey that achieved a good response rate, with a large sample of LQs. However, the education level reported was the head of households’ and this may not reflect the highest education level attainment of the family. Data were collected in 2015 and the results are likely to still reflect the current sanitation practices, as there were no drastic changes or major reform to environmental sanitation programmes in Malaysia. 

## 5. Conclusions

In general, the water and toilet facilities in Malaysia’s households were good, while the household domestic waste management practices can be improved. Despite this, there remain pockets of communities that pose environmental challenges for the nation. Thus, enabling mechanisms for good sanitation practices, improving access to improved water supply and proper waste disposal services among the specific housing types identified, as well as increasing education, should continue to be emphasised to enable infection spread prevention and improved public health.

## Figures and Tables

**Figure 1 ijerph-17-07933-f001:**
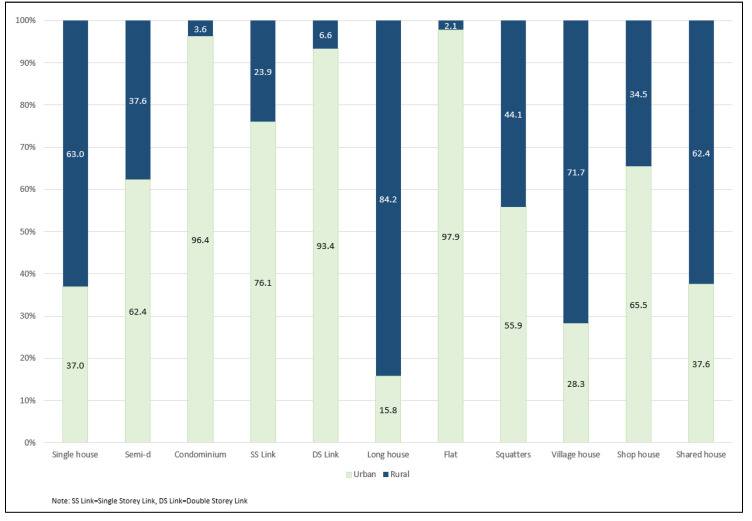
Proportion of house type by geographical location.

**Figure 2 ijerph-17-07933-f002:**
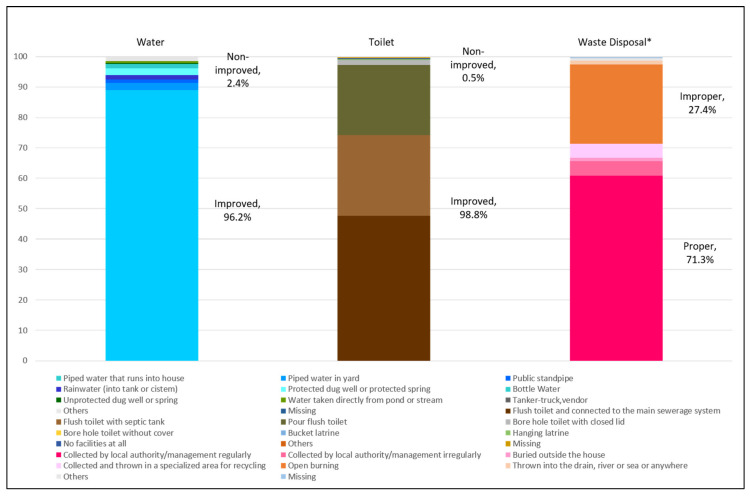
Distribution of improved and non-improved water sources, toilet types and domestic waste disposal.

**Figure 3 ijerph-17-07933-f003:**
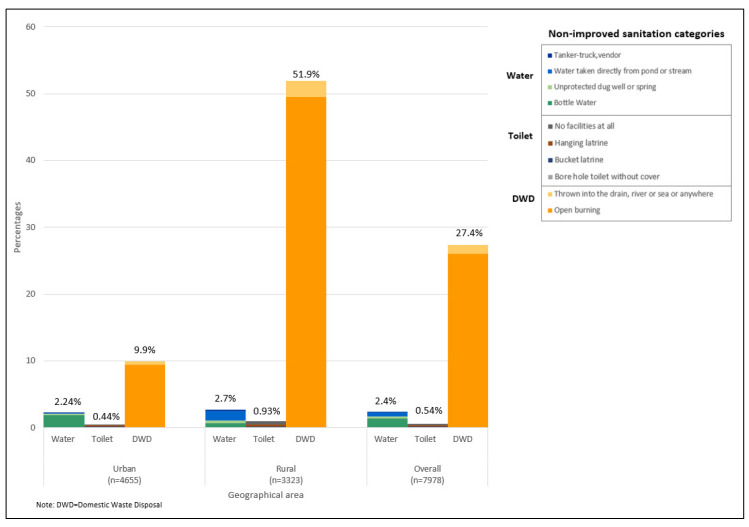
Non-improved sanitation categories by geographical location.

**Figure 4 ijerph-17-07933-f004:**
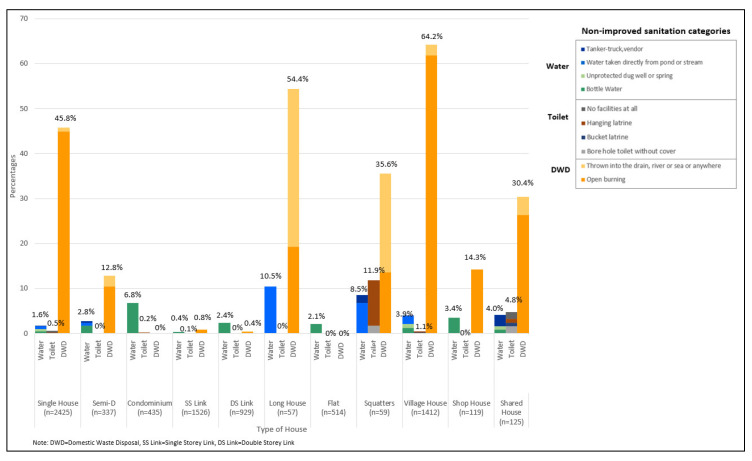
Sanitation categories by house type.

**Figure 5 ijerph-17-07933-f005:**
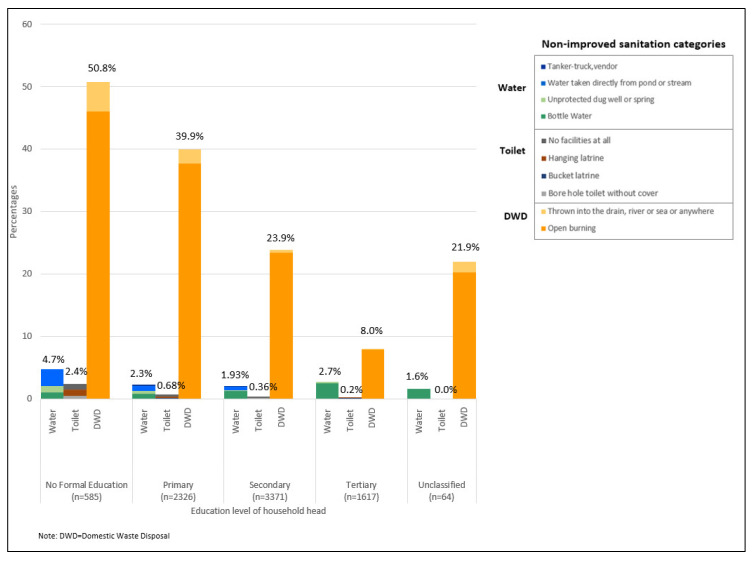
Non-improved sanitation categories by education.

**Figure 6 ijerph-17-07933-f006:**
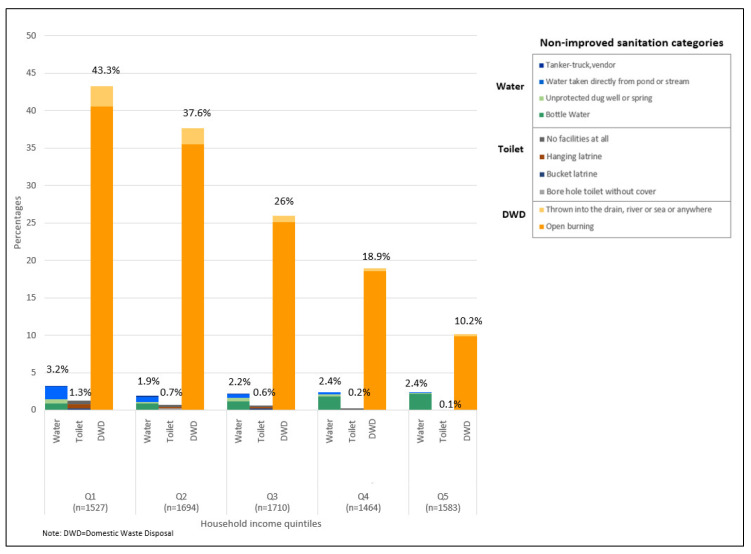
Non-improved sanitation categories by household income quintile.

**Table 1 ijerph-17-07933-t001:** Odds ratio of using non-improved drinking water sources, toilet types and improper domestic waste disposal.

Variables	Non-Improved Water		Non-Improved Toilet		Improper Domestic Waste Disposal	
	OR ^a^ (95% CI: LL, UL) ^b^	*p*-Value	OR ^a^ (95% CI: LL, UL) ^b^	*p*-Value	OR ^a^ (95% CI: LL, UL) ^b^	*p*-Value
**Education level**						
No formal	1.38 (0.79, 2.41)	0.22	2.97 (0.61, 14.41)	0.10	2.88 (2.12, 3.91)	0.00
Primary	0.69 (0.44, 1.09)	0.13	1.03 (0.22, 4.82)	0.76	1.98 (1.55, 2.53)	0.00
Secondary	0.65 (0.44, 0.98)	0.05	0.92 (0.2, 4.36)	0.87	1.7 (1.34, 2.16)	0.00
Tertiary(ref) ^c^						
**Household income quintiles**						
Q1	1.08 (0.7, 1.65)	0.67	6.00 (1.65, 21.75)	0.01	2.03 (1.68, 2.46)	0.00
Q2	0.78 (0.5, 1.21)	0.25	3.2 (0.85, 12.03)	0.10	1.78 (1.49, 2.13)	0.00
Q3	0.88 (0.58, 1.33)	0.52	3.88 (1.04, 14.5)	0.05	1.33 (1.11, 1.6)	0.00
Q4/Q5 (ref) ^c^						
**Geographical location**						
Urban (ref) ^c^						
Rural	1.22 (0.86, 1.72)	0.32	1.16 (0.59, 2.31)	0.66	3.74 (3.26, 4.29)	0.00
**House type**						
Single house (ref) ^c^						
Squatters	6.38 (2.39, 17.04)	0.00	24.86 (9.15, 67.52)	0.00	0.66 (0.37, 1.2)	0.26
Village house	2.35 (1.54, 3.58)	0.00	1.66 (0.79, 3.49)	0.18	1.80 (1.56, 2.09)	0.00
Shop house/Share house	2.19 (1.01, 4.77)	0.02	4.67 (1.72, 12.66)	0.00	0.35 (0.25, 0.49)	0.00
Others	1.45 (0.95, 2.22)	0.09	0.07 (0.01, 0.58)	0.01	0.05 (0.04, 0.07)	0.00

Note: ^a^ ORs and 95% CI adjusted for all variables in the model using binary logistic regression, the enter method. ^b^ LL = Lower Limit, UL = Upper Limit. ^c^ Ref = Reference group

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
