# Peer review of "Socio-Economic Factors Related to Drinking Water Source and Sanitation in Malaysia"

_ijerph, 2020, doi:10.3390/ijerph17217933_

Round 1

Reviewer 1 Report

This study analyzed the data from a nationwide cross-sectional survey to find the socio-economic factors related to drinking water source and sanitation in Malaysia. Data analysis are straightforward. However, some Minor modifications are necessary for this paper to be accepted for publication.

  • The part of introduction may be adjusted and more concise. It will be better to add the literature review about socio-economic factors related to drinking water source and sanitation, and reduce the discussion about importance of drinking water source and sanitation.
  • In abstract, “There is a need to increase public awareness, better education and public sanitation services to improve sanitation in the country.” but in the discussion this part are not clear, no results showed that increase public awareness better than education and public sanitation services. The concluding statement should be more rigorous.
  • Income quintile has more practical reference significance if it has specific scope.

Author Response

Dear Reviewer,

Re: Manuscript ijerph-926429

Thank you for the opportunity to revise and resubmit our manuscript titled “Socio-Economic Factors Related to Drinking Water Source and Sanitation in Malaysia”. We have provided a point-by-point response to the reviewer’s comments, and our proposed changes to the manuscript. We have also carefully re-read the manuscript and made further improvements throughout the manuscript.  

Yours sincerely,

Kong Yuke Lin, on behalf of all authors

Reviewer 2 Report

Manuscript ijerph-926429 reports an interesting study about the evaluation of drinking water managment and its sanitation in Malays. Manuscript is well structured, the methodologies used are well explained, the results are well elaborated and the conclusions acceptable. However, it needs a thorough revision by native English. 

Author Response

Dear Reviewer,

Re: Manuscript ijerph-926429

We thank the reviewer for her thoughtful review of our work and kind words. We have thoroughly re-reviewed the manuscript and corrected any errors we came across.

Yours sincerely,

Kong Yuke Lin, on behalf of all authors

Reviewer 3 Report

The authors described the drinking water supply and sanitation practices in Malaysia. In general, the study is useful and the manuscript has a good start, give a clear picture on the water quality and toilet facilities in Malaysian households. However, there are some major concerns in this manuscript:

  1. The results part included several large tables which have many information or key data, but the author did not present well in the results part. The audients may miss many key information by looking through those tables. I would recommend rewriting this part.
  2. The discussion part is more like a repeat of results, so the author should spend more time in writing the discussion. There are many similar papers, I would suggest the author read all those papers and learn how to write discussion.

Such as: Socioeconomic Factors Affecting Water Access in Rural Areas of Low and Middle Income Countries; Effect of Socio-economic Factors on Access to Improved Water Sources and Basic Sanitation in Bomet Municipality; A Spatio-Temporal Pattern and Socio-Economic Factors Analysis of Improved Sanitation in China, 2006–2015; Barriers to access improved water and sanitation in poor peri-urban settlements of Abidjan, Coˆte d’Ivoire

  1. All those tables are really hard to read, I would suggest the authors reorganized the results, plot some major results in graphs to help audients to better understand the manuscript, such as the significant correlations of each factor.
  2. A few other comments related to the introduction, line 40-42: the two numbers are not accurate according to the citation. Please check

Author Response

(The authors gave the same response as above.)
